# Vancomycin Sequestration in ST Filters: An In Vitro Study

**DOI:** 10.3390/antibiotics12030620

**Published:** 2023-03-21

**Authors:** Frédéric J. Baud, Pascal Houzé, Jean-Herlé Raphalen, Pascal Philippe, Lionel Lamhaut

**Affiliations:** 1Département d’Anesthésie et de Réanimation, Adult Intensive Care Unit, Necker Hospital, 75015 Paris, France; 2EA7323, Université de Paris, 75006 Paris, France; 3CNRS UMR 8258—U1022, Laboratoire de Biochimie, Necker Hospital, 75015 Paris, France

**Keywords:** continuous renal replacement therapy, vancomycin, dialysis, filtration, sequestration, adsorption, pharmacokinetics

## Abstract

Background. Sequestration of vancomycin in ST^®^ filters used in continuous renal therapy is a pending question. Direct vancomycin-ST^®^ interaction was assessed using the in vitro NeckEpur^®^ technology. Method. ST150^®^ filter and Prismaflex dialyzer, Baxter-Gambro, were used. Two modes were assessed in duplicate: (i) continuous diafiltration (CDF): 4 L/h, (ii) continuous dialysis (CD): 2.5 L/h post-filtration. Results. The mean initial vancomycin concentration in the central compartment (CC) was 51.4 +/− 5.0 mg/L. The mean percentage eliminated from the CC over 6 h was 91 +/− 4%. The mean clearances from the CC by CDF and CD were 2.8 and 1.9 L/h, respectively. The mean clearances assessed using cumulative effluents were 4.4 and 2.2 L/h, respectively. The mean percentages of the initial dose eliminated in the effluents from the CC by CDF and CD were 114 and 108% with no detectable sequestration of vancomycin in both modes of elimination. Discussion. Significant sequestration adds a clearance to that provided by CDF and CD. The study provides multiple evidence from the CC, the filter, and the effluents of the lack of an increase in total clearance in comparison with the flow rates without significant sequestration in the ST^®^ filter comparing cumulative effluents to the initial dose in the CC. Conclusions. There is no evidence ST^®^ filters directly sequestrate vancomycin.

## 1. Introduction

Sequestration of vancomycin in filters used in continuous renal therapy is a pending question. The first evidence of vancomycin sequestration was provided in 2008 via an in vitro study evidencing sequestration of vancomycin in polyacrylonitrile filters [1]. The amount sequestrated in the polyacrylonitrile filter was about twice that reported for the two other filters. As vancomycin remains a first choice antibiotic to treat suspected methicillin-resistant *Staphylococcus aureus* infections in critically ill patients, we considered it relevant to assess vancomycin sequestration in a new polyacrylonitrile filter used presently. As a matter of fact, in early 2022, Honore et al. suggested clinicians should be aware of the specific adsorptive properties of the membrane used during continuous renal replacement therapy (CRRT) and of the need for vancomycin monitoring [2].

Pharmacokinetics of vancomycin during CRRT has been a concern for a while. As soon as 1986, Matzke et al. performed an in vitro study dealing with the elimination of vancomycin using a polyamide filter, FH202, Gambro, in the filtration mode [3]. There was a significant linear correlation of vancomycin clearance and ultrafiltration flow rate. This study did not show any evidence of sequestration of vancomycin in a polyamide filter. Noteworthy, high-flux dialysis membranes have addressed the limitation of vancomycin elimination by dialysis in relation to its molecular weight, 1448 Da [4,5,6].

In 1992, Quale et al. studied the in vitro elimination of vancomycin at a concentration of 80 mg/L in a saline solution using cellulose and AN69 polyacrylonitrile dialyzers with the same surface area of 1.3 m^2^ [7]. Ultrafiltrate can be obtained with the polyacrylonitrile but not the cellulose filter. Vancomycin concentrations in the circuit at the first and second passage in the cellulose filter were 97 (+/− 4.5)% and 90 (+/− 9.5)%, respectively. In comparison, in the polyacrylonitrile filter, the remaining concentrations were 55 (+/− 4.9)% and 51 (+/− 0.78)%, respectively. Following 90 min of continuous recirculation through the cellulose and polyacrylonitrile filters, the vancomycin concentrations were 91 (+/− 6.8)% and 41 (+/− 0.21)%, respectively. Interestingly, only 11.9% of the amount of vancomycin removed during the 90 min of recirculation in the polyacrylonitrile filter was recovered in the ultrafiltrate. The authors concluded that the large amount of lost vancomycin was apparently bound to the polyacrylonitrile filter, which in their view, carries a negative charge [7]. 

The recent development of an in vitro model mimicking real conditions of use of the drug [8,9,10] invited us to study the direct interaction between vancomycin and a new polyacrylonitrile-derived filter of the ST^®^ series by means of NeckEpur^®^ technology. 

## 2. Results

### 2.1. Stability Studies

All the reported in vitro methods provide indirect evidence of sequestration based on the mass balance principle during the study period. In the present study, we tested a mean concentration of 51 mg/L, which is slightly above the upper limit of the therapeutic range. Stability was assessed in the reservoir (central compartment) and in the tubes used for the samplings.

#### 2.1.1. In the Sampling Tubes

At +6 h, the mean percentage of variation was −2.6%.

#### 2.1.2. In the Reservoir (Central Compartment)

At +6 h, the mean percentage of variation was −1.2%.

Stability of the drug in in vitro studies is a pre-requisite prior to any further assessment. Our results allow us to go further. 

### 2.2. Actual Flow Rate Provided by the Dialyzer

In CRRT, the dialysis/filtration flow rate is a major determinant of drug clearance. Therefore, one major issue is to assess the correlation between the prescribed and the actual delivered dialysis/filtration flow rate. 

During the two continuous diafiltration (CDF) sessions at a 4 L/h flow rate, the theoretical time for collection of a 5-L bag is 75 min. The mean measured collection of a 5-L bag was 76 + 9 min.

During the two continuous dialysis (CD) sessions at a 2.5 L/h flow rate, the theoretical time for collection of a 5-L bag is 120 min. The mean measured time of collection was 123 + 1 min.

Our data show a close agreement between the prescribed and the actual delivered dialysis/filtration flow rate.

### 2.3. Elimination from the Central Compartment (CC)

The mean initial concentration of vancomycin was 51.4 +/− 5.0 mg/L. During the 6-h study period, the mean percentage of vancomycin eliminated from the CC by the different routes of elimination was 91.4 +/− 4.0%. The nearly complete elimination of vancomycin is required for providing definitive conclusions.

#### 2.3.1. In the Continuous Diafiltration Mode

The mean concentration of vancomycin in the CC was 54.0 mg/L.

The mean AUC in the CC was 5599 mg.min.L^−1^. The mean clearance from the CC was 2.7 L/h. Due to loss of instantaneous effluent samplings, clearance from the effluents was calculated in only one session. Clearance from the effluents was 4.4 L/h. The mean EC was 9%.

The mean value of the percentage of elimination resulting from diafiltration and sequestration were 114 and 0%, respectively.

Figure 1 shows the time courses of vancomycin concentrations in the CC, inlet, outlet, and instantaneous effluent ports in the two sessions. Due to loss of instantaneous effluent samplings in CDF-2, data of instantaneous effluents are presented in CDF-1 only.

#### 2.3.2. In the Continuous Dialysis CD Mode

The mean measured initial concentration of vancomycin in the CC was 48.9 mg/L.

The mean AUC in the CC was 6930 mg.min.L^−1^. The mean clearance from the CC was 1.9 L/h. The mean clearance calculated from the effluent was 2.2 L/h. The mean EC was 14%.

The mean value of the percentage of elimination resulting from diafiltration and sequestration were and 108 and 0%, respectively.

Table 1 shows individual data collected in the two sessions.

Figure 2 shows the time courses of vancomycin concentrations in the CC, inlet, outlet, and instantaneous effluent ports in the two sessions in the CD mode.

### 2.4. Summary of Results

Elimination of vancomycin was assessed in the CDF mode at a 4 L/h flow rate and the CD mode at a 2.5 L/h. Our results consistently showed the lack of sequestration.

## 3. Discussion

In 2008, Tian et al. compared the adsorption of vancomycin by polyacrylonitrile, polyamide, and polysulfone filters [1] using a closed one-compartment model with hemofiltration returning to the reservoir. The surface area of the polyacrylonitrile and polyamide filters was 0.6 m^2^. Vancomycin was added to blood mixed with a crystalloid solution. Heparin 5000 IU was added to diluted blood in the circulating session. The extents of adsorption of vancomycin in the polyacrylonitrile and polyamide filters were 10.08 +/− 2.26 mg and 5.20 +/− 1.82 mg, respectively, that should be compared to the initial 36 mg dose of vancomycin. Cumulative adsorption was not changed by the addition of 500-mL Ringer’s solution to reduce the circulating vancomycin concentration, suggesting irreversible adsorption. The authors mentioned that the AN69 polyacrylonitrile filter carries a negative charge, whereas vancomycin carries a positive charge allowing ionic drug-filter interaction to occur. On the contrary, in 2012, using two polysulfone filters, Pinner et al. performed an in vitro study, whose results did not suggest any sequestration of vancomycin [11].

In 2014, Jang et al. performed an in vitro study using an open model of a hemodialysis system with a separate collection of dialysis effluent; 30 mg/L of vancomycin was added to diluted/reconstituted blood. The use of anticoagulant was not specified. Polyflux^®^ Gambro (chemical structure not cited) and OptiFlux^®^ Fresenius, a polysulfone filter, were used. Total vancomycin recovery in the dialysis effluents was 85 +/− 18%, suggesting that up to 15% may have been sequestrated in the dialysis filter or tubing [12].

In 2020, in their first in vitro study, Onichimowski et al. assessed the adsorption of vancomycin in a new polyacrylonitrile-derived Gambro filter of the ST^®^ series [13]. In the study, 1000 mg of vancomycin was diluted in a 5-L bag of buffered crystalloid solution. The ultrafiltration rate was set to 0.5 L/h and the fluid flow rate to 100 mL/min, using a closed one-compartment model with the filtration line returning to the central reservoir. Seven samples were collected at the post-filter site. The authors concluded, using only the initial concentration and the concentrations at +120 min, that adsorption of vancomycin in the AN69-ST^®^ filter was 181.88 mg (i.e., 18.2%).

In another in vitro study, Onichimowski et al. studied the adsorption of vancomycin in the ST^®^ filter, but using porcine blood [14]. The same closed one-compartment model was used with the ultrafiltrate returning to the reservoir. The initial 1000-mg dose of vancomycin was added to 1010 mL of blood. Accordingly, the initial theoretical concentration, 990 mg/L, was about five-fold greater than that in the previous study. Eight samples were collected at the pre-filter site. There was a significant drop of concentration maximal at +15 min. Thereafter, the level of the drug in the blood returned to about 80% of the initial values, suggesting vancomycin was released from the filter. The authors reported that 195.26 mg (19.7%) of vancomycin was adsorbed in the ST filter.

In the in vitro studies performed by Onichimowski et al., the respective initial concentrations of 200 and 990 mg/L that were used are far in excess when compared to the range of therapeutic values of plasma vancomycin concentrations. Therefore, the two studies deal with toxic rather than therapeutic concentrations. The safety margin of vancomycin is low, and too-high concentrations expose patients to the risk of red man syndrome (RMS), known to be the most common adverse reaction to vancomycin [15]; RMS is an anaphylactoid reaction caused by the degranulation of mast cells and basophils after rapid infusion of vancomycin, resulting in the release of histamine. In our study and that of Matzke et al. [3], an initial concentration of 50 mg/L, slightly above the upper limit of the therapeutic range, was studied.

In vitro studies performed to assess vancomycin adsorption in filters highlight the concern of polyacrylonitrile filters. Indeed, the AN69 filters used by Quale et al. and Tian et al., which are made of polyacrylonitrile, are strongly electronegative, whereas the ST^®^ filters used by Onichimowski et al. in their two in vitro studies, and in our in vitro study are far less electronegative. Indeed, polyacrylonitrile is covered with polyethylene imine (PEI), which significantly decreased the negative charge of the filter to increase its biocompatibility [13]. Therefore, it is of utmost importance to not extend the results obtained with vancomycin sequestration in the AN-69 filter [1,7] to those with the ST^®^ filter covered with PEI and collected by Onichimoski et al. [14] and our present study. Furthermore, the 0.6 m^2^ surface area of filters used in Tian’s study fits well with studies in the pediatric population, whereas the (1.7 m^2^ and 1.5 m^2^) surface areas of the filters used by Quale et al., Onichimowski et al., and our present study fit well with studies performed in the adult population. The impact of the surface area on the extent of sequestration has been consistently reported [5,16,17].

The studies dealing with in vitro assessment of drug sequestration in filters, including the present study, highlight major differences in the methodologies used to investigate the phenomenon. Regarding the study design, Tian et al. and Onichimowski et al. used a closed model with the ultrafiltrate returning to the reservoir, whereas, as Matze and Quale, we used an open model with the ultrafiltrate collected apart, providing instantaneous and cumulative effluents. In the closed model, the only way for the drug to escape is to be adsorbed by the filter and the circuit tubing. This model may overestimate drug adsorption. In contrast, the separate collection of effluents allows for a comparison of the elimination of drugs from the reservoir by dialysis/filtration to that sequestrated in the filter [3,7]. The latter model mimics more closely the condition of a session of dialysis/filtration in humans than the closed model. Furthermore, Tian et al. [1] and Onichimowski et al. [13,14] collected samplings at only one site of the whole circuit, whereas Matzke et al. and Quale et al. [3,7] collected samples at the inlet and outlet ports of the filter, as we did in the present study. Noteworthy, Kronfol et al. evidenced in vitro aminoglycoside sequestration in a polyacrylonitrile filter by means of repeatedly collecting samples at the inlet and outlet ports of the filter [16]. In the general principles of pharmacokinetics, the effect of an organ or a device on drug pharmacokinetics is studied by means of collecting data on concentrations and flow rates at their inlet and outlet ports [18]. We do believe that in vitro studies assessing drug sequestration in filters, whatever the filter used [19], should comply with the general principles of pharmacokinetics [20].

To assess further the differences between the in vitro models, there is a need to consider the medium used to assess drug adsorption. Reconstituted/diluted blood was used in four studies [1,3,12,14], whereas crystalloid solution was used in two previously published studies [7,13] to which our study should be added. The medium used defines the problem being addressed. Using a crystalloid solution allows the assessment of the drug-filter interaction, whereas using reconstituted/diluted blood allows assessment of the drug-blood-filter interaction. Blood is nothing but a complex living liquid tissue inducing drug-blood interactions, including protein binding and blood-to-plasma distribution. Protein binding is a well-known factor resulting in restriction of drug distribution within the body, acting through a decrease in the unbound fraction. Similarly, protein binding restricts access of a drug to a filter. As a matter of fact, the rate of drug sequestration studied in protein-containing media, including whole blood, plasma, and albumin, is consistently lower low than that assessed in a crystalloid medium [21]. Another masked effect of whole blood includes drug distribution into erythrocytes defined by the blood-to-plasma ratio (Kp_BC_) [22]. Spiking whole blood instead of plasma increases the need for awareness of the blood-to-plasma ratio. The determination of Kp_BC_ requires the measurement of hematocrit (Hte), the blood-to-plasma concentration ratio (C_b_/C), and the fraction of the drug in plasma unbound (fu) to proteins, according to the following equation:C_b_/C= 1 + Hte (fu × Kp_BC_ − 1)

Assuming for vancomycin a Kp_BC_ of 0.55, a measured Hte of 33.6%, and an unbound vancomycin fraction of 55% [23], the blood-to-plasma ratio for vancomycin in Onichimowski’s study [14] would have been 0.75. According to the Kp_BC_ value, 1000 mg of vancomycin in 1010 mL with a hematocrit at 33.6% would have resulted in 885 mg of vancomycin in 671 mL of plasma (1319 mg/L) and 115 mg of vancomycin in a total erythrocyte volume of 339 mL (339 mg/L). The distribution of vancomycin into erythrocytes would have accounted for a 13% decrease in the amount of vancomycin in plasma, not related to filter sequestration. Furthermore, blood includes a cellular compartment having a limited lifespan when drawn out of the body. Spontaneous hemolysis of collected blood is a major concern when measuring several biochemical parameters, including blood gases and kalemia. Hemolysis may be assessed by many methods, including discoloration of plasma after centrifugation of blood, measurement of kalemia, lactate dehydrogenase activity, free plasma hemoglobin concentration, serum haptoglobin concentration, or even the carboxyhemoglobin level [24]. Assessing the blood-to-plasma ratio, Yu et al. reported that, in some experiments requiring incubation at 37 °C of whole blood for a period beyond 1 h, hemolysis occurred and was evidenced by a pink or reddish discoloration of plasma, which resulted in highly variable results [25]. Furthermore, hemolysis may result in significant pharmacokinetic alterations, including extending the plasma volume with the erythrocyte volume with a lower concentration of vancomycin due to the low blood-to-plasma ratio, as shown above, which would further decrease the plasma concentration along the session. As sessions of in vitro studies using blood lasted 2 h [13], 3.5 h [3], and 4 h [1], there was a high risk of significant hemolysis. We are not aware of any study having reported information to assess the extent of hemolysis during sessions lasting several hours at 37 °C.

Using a crystalloid solution provides insight on the pharmacokinetics of the unbound fraction of vancomycin and the direct vancomycin filter interaction. Crystalloid solution obviates the need for anticoagulation and consideration for hemolysis. Matzke et al. did not specify the anticoagulant they used, while Onichimowski et al. used citrate. In contrast, Tian added 5000 IU/L of heparin in the CC. Heparin binds several drugs including vancomycin [26]. Whether heparin may have resulted in decreasing plasma concentrations of vancomycin in Tian’s study remains to be determined.

Using the same ST^®^ filter and a crystalloid medium, there is some apparent discrepancy between Onichimowski’s study [13] and our present study. The time-course of concentrations collected at the pre- and post-filter ports in the CDF and CD modes did not reveal any significant decrease in vancomycin concentrations, which could suggest vancomycin sequestration. Noteworthy, significant drug sequestration in the filter alters the time course of EC values, with initial high values decreasing to expected values within 120 min. This finding was observed in both the filtration mode [16] as well as the dialysis and diafiltration modes [9]. This finding was not reported in the two Onichimowski’s studies, as samples were collected either from the pre- or post-filter ports but never at both ports simultaneously. Onichimowski et al. reported a rebound of concentrations not significantly different from the control from 45 to 90 min, suggestive of reversible sequestration [14]. We are not aware of any irreversible sequestration having started so late after initiation of the session and preceded by a phase of reversible sequestration. At the end of the two studies, statistically significant differences in concentrations were observed using Student’s *t*-test. However, the design of the two studies is fitting criteria for analysis by means of two-way ANOVA for repeated measurements. Among seven and eight couples of data, comparison of only the first and the last point is questionable. In studies using an extremely low number of data, the use of parametric statistical test is questionable, whereas non-parametric tests would provide more robust analysis. In the condition of an extremely low number of data, Phillips et al. proposed a 17% threshold value for the difference between values [27]. In an in vitro study dealing with vancomycin, Jang et al. proposed a 15% threshold [12]. The US FDA recommends inter-assay reproducibility of bioanalysis being less than or equal to 15% [28]. In in vitro studies, we proposed a 20% threshold, which seems relevant when comparing with clinical conditions [8,9,29,30]. Compared to the 1000 mg of vancomycin added in the circuit, Onichimowski et al. reported sequestrated amounts in the crystalloid and blood studies of 181.88 mg and 195.26 mg, respectively. These amounts correspond to eliminated percentages lower than or nearly equal to the 20% threshold of clinical significance. Our study lasted 6 h in comparison with 2 h in both Onichimowski’s studies. In contrast with Onichimowski study calculating instantaneous clearances, in our study, total clearance was calculated using two independent sets of data: on one hand, areas under the curve (AUCs) from the CC, and on the other hand, AUCs from instantaneous effluents. Both data sets did not provide evidence of any additional terms of clearance to data provided by diafiltration and dialysis. The hallmark of significant sequestration is a total clearance greater than that provided by diafiltration or dialysis that is strongly correlated to the flow rate when using a crystalloid solution in the absence of sequestration [10]. Our results in the different modes did not show any increase in the total clearance of vancomycin whatever the methods used to calculate clearance, CDF or CD. Finally, in the four sessions of our study, the comparison of the initial amounts in the CC to the cumulative amounts eliminated in the effluents consistently showed extensive elimination by diafiltration as well as dialysis.

Our study suffers from several limitations. We studied only one single concentration, while the range of therapeutic concentrations might also be considered. However, we tested a concentration slightly above the upper limit of the therapeutic range. We studied sequestration using only two flow rates, 2.5 and 4 L/h, while Onichimowski et al. studied flow rates of 0.5 and 0.6 L/h. However, in the adult population, a filtration rate of 0.5–0.6 L/h is rarely used. The CF mode had been the recommended mode of CRRT for elimination of vancomycin. The CF mode was not investigated. However, our study confirms that high-flux filters including the ST^®^ series efficiently eliminate vancomycin in the dialysis mode. In contrast with Onichimowski et al., we did not assess the concentrations at 5 min. However, it seems unlikely that a missing sample at 5 min in our study may have significantly altered the conclusion. Our study shows the lack of direct vancomycin-ST interaction, which does not exclude indirect sequestration due to the binding of vancomycin to proteins and protein sequestration in filters. The binding of vancomycin to proteins ranges from 11 to 55% [5], which is considered as mild to moderate. Furthermore, the binding of drugs to proteins is reversible, and the consequence of indirect sequestration on the overall vancomycin pharmacokinetics remains to be determined.

Another major limitation results from the mode of exposure of the filter to the drug. All in vitro studies, including the present one, have been performed using a bolus dose, i.e., exposure of the filter to the initial maximal concentration. However, this mode of exposure does not meet that of vancomycin in clinical practice. Indeed, an in vitro study should be designed to address the elimination of a therapeutic concentration, as we did. Conversely, other in vitro studies are designed to address the elimination of a therapeutic dose, as Onichimowski et al. did [13,14]. This design exposes the filter to extremely toxic concentrations that are never expected to occur in clinical practice. The future of in vitro studies is to propose models addressing the actual conditions of use of vancomycin based on a continuous intravenous infusion. We believe our results support the assumption that vancomycin monitoring allows efficient dose adaptation to targeted plasma concentrations when using ST^®^ filters.

## 4. Materials and Methods

### 4.1. Vancomycin Preparation

We used the specialty currently used in our ICU department: 1000 mg of vancomycin per vial Mylan^®^ was supplied by the pharmacy of our institution, Hospital Necker.

### 4.2. Stability Study

Prior to the studies, stabilities of amikacin in Vacutainer^®^ tubes used for sampling and the 5-L bag of Hemosol^®^ used as the central compartment were studied at ambient temperature just after injection and at +6 h post-injection.

### 4.3. Measurements of Vancomycin Concentrations

Vancomycin concentrations were determined using the immunochemical method, linear from 2 (limit of quantification) to 50 mg/L, (clinical analyzer C-8000 (Abbott, France)).

### 4.4. Filter and Dialyzer Device

The Prismaflex^®^ diafiltrator (Baxter-Gambro, Paris, France) and the ST^®^150 filter were used. A new filter was used for each session.

### 4.5. Modes of Exposure of the Filter to Vancomycin

As previously reported [8,9,10,29,31], the 5-L bag of the CC was loaded with vancomycin at the targeted concentration of 50 mg/L just prior to initiation of the session. This mode of exposure corresponds to exposure of the CC to the maximal concentration which defines a bolus mode. The crystalloid solute used in the different compartments was Hemosol^®^ [29].

### 4.6. Mode of CRRT

We studied continuous dialysis (CD) and continuous diafiltration modes (CDF). Each session was made in duplicate. The simulated blood flow rate was set to 200 mL/min. In the CDF mode, the dialysis flow rate was 4 L/h, combining a dialysis flow rate set to 2500 mL/min with prefiltration and post-dilution flow rates set to 500 and 1000 mL/min, respectively. The CD flow rate was set to 2500 mL/h. The net ultrafiltration flow rate was set at zero.

The sessions were limited to 6 h, resulting in the elimination of more than 90% of the initial amount in the CC in the CDF mode.

### 4.7. Samplings

Samples were collected in the CC, at the inlet (C_si_), outlet (C_so_), effluent line for assessment instantaneous effluent concentration (C_efflu instant_), and in the filled bag for collection of the cumulative effluent (C_effl cum_). Samples were taken at 0, +15 min, +30 min, +45 min, +60 min, and then at + 2, 3, 4, and 6 h. Cumulative effluents were sampled at the time the bag was filled out and needed to be changed.

### 4.8. Assessment of the Routes of Elimination

The routes of elimination were studied using the NeckEpur_®_ technology, as previously reported [29]. As recommended [32], the extraction coefficient (EC) was calculated in the different modes of CRRT as the ratio of (C_si_- C_so_)/C_si_ expressed as a percentage. The clearance (CL) from the central compartment over the study period was calculated as follows: CL = Dose/AUC, using repeated measurements of the concentrations in the central compartment [29,33]. Clearance was also calculated according to the total amount eliminated in effluents, applying the flowing equation: CL = Dose/AUC, where AUC means the area under the curve of instantaneous effluent concentrations.

### 4.9. Actual Flow Rate Provided by Prismaflex Dialyzer

The accuracy of the actual flow rates provided by the Prismaflex over a 6-h session was assessed by measuring the times needed to fill 5-L bags. The bags were changed when indicated by the corresponding alarm. The expected durations of filling at 4.0 L/h and 2.5 L/h were 75 min and 120 min, respectively.

### 4.10. Statistical Analysis

Owing to the small number of experiments, comparisons were conducted as previously reported [29,30]. A difference in the means greater than or equal to 20% was considered clinically relevant. For data collected in duplicate or less, results are presented as means only. For data collected in more than duplicate (mean concentration, EC), results are presented as mean +/− SD. Statistics of columns were made using the software GraphPad Prism^®^ V9.

## 5. Conclusions

The present study provides robust evidence that vancomycin at a 50 mg/L concentration in the central compartment is not sequestrated in the ST^®^150 filter. Therefore, we conclude vancomycin does not require therapeutic adaptation more than that required by dialysis and filtration during sessions of continuous renal replacement therapy using the ST^®^ filter.

## 6. Patents

NeckEpur^®^ technology has been patented.

## Figures and Tables

**Figure 1 antibiotics-12-00620-f001:**
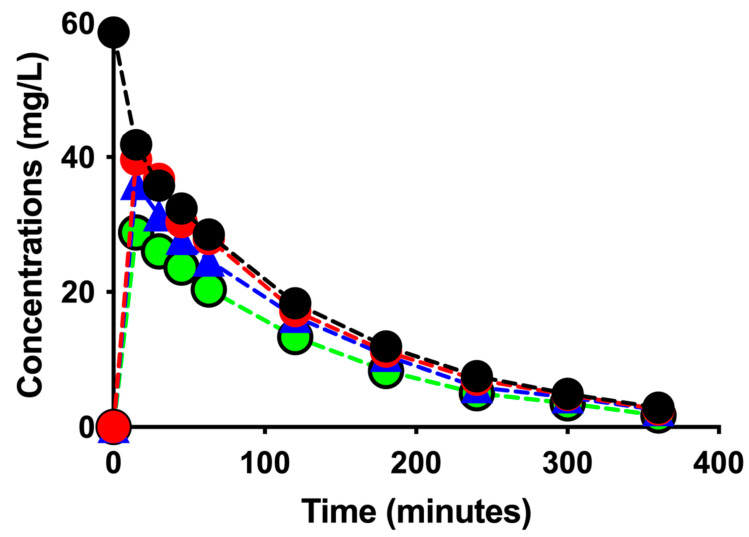
Time courses of vancomycin concentrations in the CC (black filled circles with dotted black line), inlet (red filled circles with dotted red line), outlet (blue filled triangles with dotted blue line), and instantaneous effluent ports (green filled circles with dotted green line) in the two sessions.

**Figure 2 antibiotics-12-00620-f002:**
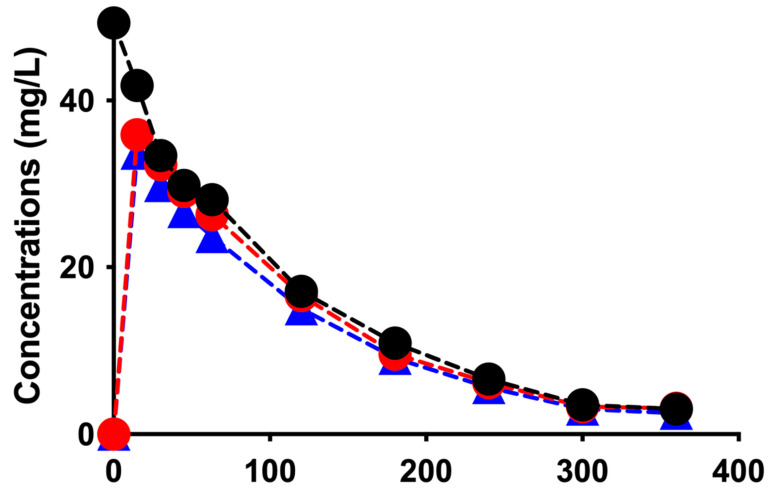
Time courses of vancomycin concentrations in the CC (black filled circles with dotted black line), inlet (red filled circles with dotted red line), outlet (blue filled triangles with dotted blue line), and instantaneous effluent ports (green filled circles with dotted green line) in the two sessions in the CD mode.

**Table 1 antibiotics-12-00620-t001:** Pharmacokinetic parameters of vancomycin elimination from the CC in each session in the CDF and CD modes by the ST^®^ filter.

Sessions/Pharmacokinetic Parameters	CDF *		CD **	
	CDF-1	CDF-2	CD-1	CD-2
Concentrations in the CC (mg/L):				
T0 min	58.6	49.3	50.7	47.0
T360 min	2.9	3.0	6.0	6.1
AUC_0–360_ (mg.min/L)				
CC	5824	5375	6968	6893
Instantaneous effluents	3814	-	5833	5003
Extraction coefficient (%)	10 +/− 4	9 +/− 4	12 +/− 3	15 +/− 1
% Eliminated from the CC	95.1	93.9	86.2	87.0
Clearance from CC (L/h)	2.41	2.34	2.18	2.05
Clearance calculated from AUC_efflu instant_ (L/h)	4.4	-	1.9	2.5
% Eliminated by CDF or CD	114	114	113	102
% Sequestrated	0	0	0	0

* Denotes continuous diafiltration. ** Denotes continuous dialysis. CC denotes central compartment. AUC denotes area under the curve.

## Data Availability

Data supporting reported results can be found at the Department of anesthesiology and intensive Care, Hôpital Necker, 149, rue de Sèvres. 75015 Paris, France.

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
