# Peer review of "Vancomycin Sequestration in ST Filters: An In Vitro Study"

_antibiotics, 2023, doi:10.3390/antibiotics12030620_

Round 1
Reviewer 1 Report
The article is important research given the increased use of renal replacement therapy and vancomycin in medicine. I have some comments for improvement. Please see my comments for improvement
1- Please avoid declarative titles. I suggest you change the title to Vancomycin sequestration in ST filter: An vitro study
2- Please make the abstract structured based on the journal style: background, method, results, conclusion
3- Make sure bacteria names italicized
4- The introduction should be expanded. I suggest you add a paragraph that should be the first paragraph for statistics of hemodialysis and vancomycin used. I think there are a lot of statistics. This would make your article better for readers
5- Paragraph in the discussion starting with “On the contrary, in 2012, using two polysulfon” I don’t like how this paragraph look only 2 lines? Can you try to merge it with other paragraphs? This will be better
Author Response
1- Reviewer: "Please avoid declarative titles. I suggest you change the title to Vancomycin sequestration in ST filter: An vitro study"
Answer: We agree with the reviewer's comment and changed the title in the revised version, as follows: "Vancomycin sequestration in ST filter: an vitro study"
2- Reviewer: " Please make the abstract structured based on the journal style: background, method, results, conclusion"
Answer: We structured the abstract as requested; we added a paragraph dealing with discussion prior to conclusion as follows:
"
Background. Sequestration of vancomycin in STÒ filters used in continuous renal therapy is a pending question. Direct vancomycin-STÒ interaction was assessed using the in vitro NeckEpurÒ technology. Method: ST150â filter and Prismaflex dialyzer, Baxter-Gambro were used. Two modes were assessed in duplicate: i) continuous diafiltration (CDF): 4 L/h, ii) continuous dialysis (CD): 2.5 L/h post-filtration. Results. The mean initial vancomycin concentration in the central compartment (CC) was 51.4 + 5.0 mg/L. The mean percentage eliminated from the CC over 6h was 91 + 4%. The mean clearances from the CC by CDF and CD were 2.8 and 1.9 L/h, respectively. The mean clearances assessed using cumulative effluents were 4.4 and 2.2 L/h, respectively. The mean percentages of the initial dose eliminated in the effluents from the CC by CDF and CD were 114 and 108% with no detectable sequestration of vancomycin in both modes of elimination. Discussion. Significant sequestration adds a clearance to that provided by CDF and CD. The study provides multiple evidence from the CC, the filter, and the effluents of the lack of increase of total clearance in comparison with the flow rates without significant sequestration in STâ filter comparing cumulative effluents to the initial dose in the CC. Conclusion. There is no evidence STâ filter directly sequestrates vancomycin. "
3- To address the reviewer's comment: "Make sure bacteria names italicized"
Answer: we found two names of bacteria that are italicized in the revised manuscript: Page 2-line 39 and ref 26.
4- Reviewer's comment: "The introduction should be expanded. I suggest you add a paragraph that should be the first paragraph for statistics of hemodialysis and vancomycin used. I think there are a lot of statistics. This would make your article better for readers."
Answer. We agree with the reviewer and changed the place of first paragraph for statistics of hemodialysis and vancomycin used moved the text from discussion to introduction, as follows:" (Please note, the numbering of references will be made on the final revised version).
- Introduction
Sequestration of vancomycin in filters used in continuous renal therapy is a pending question. The first evidence of vancomycin sequestration was provided in 2008 via an in vitro study evidencing sequestration of vacomycin in a polyacrylonitrile filters [1]. The amount sequestrated in the polyacrylonitrile filter was about twice those reported for the two other filters. As vancomycin remains a first choice antibiotic to treat suspected methicillin-resistant staphylococcus aureaus infections in critically ill patients, we considered relevant to assess vancomycin sequestration in anew polyacrylonitrile filter used presently. As a matter of fact, in the early 2022 Honore et al, suggested clinician should be aware of the specific adsorptive properties of the membrane used during continuous renal replacement therapy (CRRT) and of the need for vancomycin monitoring [2].
Pharmacokinetics of vancomycin during CRRT has been a concern for a while. As soon as 1986, Matzke et al performed an in vitro study dealing with the elimination of vancomycin using a polyamide filter, FH202, Gambro, in the filtration mode [6]. There was a significant linear correlation of vancomycin clearance and ultrafiltration flow rate. This study did not show any evidence of sequestration of vancomycin in a polyamide filter. Noteworthy, high-flux dialysis membranes have addressed the limitation of vancomycin elimination by dialysis in relation to its molecular weight, 1 448 Da [7-9].
In 1992, Quale et al studied the in vitro elimination of vancomycin at a concentration of 80 mg/L in a saline solution using cellulose and AN69 polyacrylonitrile dialyzers with the same surface area of 1.3 m2 [10]. Ultrafiltrate can be obtained with the polyacrylonitrile but not the cellulose filter. Vancomycin concentrations in the circuit at the first and second passage in the cellulose filter were 97(+ 4.5)% and 90(+ 9.5)%, respectively. In comparison, in the polyacrylonitrile filter, the remaining concentrations were 55(+4.9)% and 51(+ 0.78) %, respectively. Following 90 min of continuous recirculation through the cellulose and polyacrylonitrile filters, the vancomycin concentrations were 91 (+ 6.8)% and 41 (+0.21)%, respectively. Interestingly, only 11.9 % of the amount of vancomycin removed during the 90 min of recirculation in the polyacrylonitrile filter was recovered in the ultrafiltrate. The authors concluded that the large amount of lost vancomycin was apparently bound to the polyacrylonitrile filter, which in their view, carries a negative charge [10].
The recent development of an an in vitro model mimicking real conditions of use of drug [3-5] invited us to study the direct interaction between vancomycin and a new polyacrylonitrile-derived filter of the series ST® by means of NeckEpurÒ technology."
5- Reviewer's comment :" Paragraph in the discussion starting with “On the contrary, in 2012, using two polysulfon” I don’t like how this paragraph look only 2 lines? Can you try to merge it with other paragraphs? This will be better"
Answer: to address the reviewer's comment, we mege the first and second two-lines paragraph at the beginning of discussion.
Reviewer 2 Report
Considering that vancomycin remains a first choice antibiotic to treat methicillin-resistant staphylococcus infections, especially in critical ill patients, your study highlights valuable information for this group of population. For a better presentation of your results, it would be nice if you could add data related to the administered dose that presumably was adapted accordingly for dialysed patients and the time period. Additionally, below the figure and table, you should include a note explaining the used abbreviations. It appears, Figure 1 title is missing.
Author Response
1- Reviewer's comment: "Considering that vancomycin remains a first choice antibiotic to treat methicillin-resistant staphylococcus infections, especially in critical ill patients, your study highlights valuable information for this group of population. For a better presentation of your results, it would be nice if you could add data related to the administered dose that presumably was adapted accordingly for dialysed patients and the time period. "
Answer: we thank the reviewer for this challenging question. In the chapter dedicated to the limitation of our study ine lines-280-287, we added the text in the revised version, as follows:
"Another major limitation results from the mode of exposure of the filter to the drug. All in vitro studies, including the present one, have been performed using a bolus dose, i.e, exposure of the filter to the initial maximal concentration. However, this mode of exposure does meet that of vancomycin in clinical practice. Indeed, an in vitro study should be designed to address the elimination of a therapeutic concentration, as we did. Conversely, other in vitro studies are designed to address the elimination of a therapeutic dose, as Onichimowski et al did[13, 14]. This design exposes the filter to extremely toxic concentrations, never expected to occur in clinical practice. The future of in vitro studies is to propose models addressing the actual conditions of use of vancomycin based on a continuous intravenous infusion. "
2- Reviewer's comment: "Additionally, below the figure and table, you should include a note explaining the used abbreviations. It appears, Figure 1 title is missing."
Answer: all comments are addressed:
In text, as follows:
"Figure 1 shows the time-courses of vancomycin concentrations in the CC (black filled circles with dotted black line), inlet (red filled circles with dotted red line), outlet (blue filled triangles with dotted blue line), and instantaneous effluent ports (green filled circles with dotted green line) in the 2 sessions."
"Figure 2 shows the time-courses of vancomycin concentrations in the CC (black filled circles with with dotted black line), inlet (red filled circles with dotted red line), outlet (blue filled triangles with dotted blue line), and instantaneous effluent ports (green filled circles with dotted green line) in the 2 sessions in the CD mode."
At the bottom of the Table